# Effects of Combining the Genes Controlling Anthocyanin and Melanin Synthesis in the Barley Grain on Pigment Accumulation and Plant Development

Anastasiya Glagoleva [1,2,*], Tatjana Kukoeva [1,2], Sergey Mursalimov [1], Elena Khlestkina [1,3] and Olesya Shoeva [1,2]

1 Institute of Cytology and Genetics, Siberian Branch of Russian Academy of Sciences (ICG SB RAS), 630090 Novosibirsk, Russia; kukoeva@bionet.nsc.ru (T.K.); mursalimov@bionet.nsc.ru (S.M.); khlest@bionet.nsc.ru (E.K.); olesya_ter@bionet.nsc.ru (O.S.)
2 Kurchatov Genomic Center, Institute of Cytology and Genetics, Siberian Branch of Russian Academy of Sciences (ICG SB RAS), 630090 Novosibirsk, Russia
3 N.I. Vavilov All-Russian Institute of Plant Genetic Resources (VIR), 190000 St. Petersburg, Russia
* Correspondence: glagoleva@bionet.nsc.ru

**Abstract:** Anthocyanins and melanins are phenolic pigments of plants and accumulate in seed envelopes of the barley grain, thereby giving them a blue, purple, or black color. To explore the effects of combined accumulation of anthocyanins and melanins in the grain, a barley near-isogenic line (NIL), characterized by simultaneous accumulation in both pigments, was developed using a marker-assisted approach. The presence of both pigments in the grain pericarp was evaluated by light microscopy. Emergence of anthocyanin pigmentation proved to be temporally separated from that of melanin, and the formation of anthocyanin pigments began at an earlier stage of spike maturation. During spike maturation, a significantly higher total anthocyanin content was noted in the created NIL than in the parental anthocyanin-accumulating NIL, indicating a positive influence of the *Blp1* gene on the anthocyanin content at some developmental stages. In a comparative analysis of yield components, it was found that the observed differences between the barley NILs are possibly caused by environmental factors, and the presence of pigments does not decrease plant productivity. Our results should facilitate investigation into genetic mechanisms underlying overlaps in the biosynthesis of pigments and into breeding strategies aimed at the enrichment of barley varieties with polyphenols.

**Keywords:** barley; phenolic compounds; anthocyanins; melanin; overlapping metabolic pathways

## 1. Introduction

Barley (*Hordeum vulgare* L.) is one of the most cultivated grain crops in the world, ranked fourth after rice, wheat, and maize [1]. Barley is widely used as a feed for livestock, as a raw material for malt production in breweries, and as a food source in the human diet. Due to its ability to grow in a wide range of climates, the crop is essential in locations where other cereals cannot be cultivated successfully [2]. Barley's high nutritional value and stress tolerance explain increased interest in this crop as a food source. Aside from containing basic nutrients, barley whole grains are rich in dietary fiber, indoles, and phenolic compounds, which are recommended components of a healthy human diet [3,4].

Phenolic compounds found in barley such as cinnamic and benzoic acid derivatives, proanthocyanidins, flavonols, flavanones, flavones, anthocyanins, and melanins are strong antioxidants, which serve as free-radical scavengers, reducers of pro-oxidant metals, and quenchers of singlet oxygen formation [5–7]. These compounds are secondary metabolites and are involved in plant growth and protection from unfavorable conditions such as the cold, drought, UV radiation, and pathogen invasion [8–11]. It has been demonstrated that in addition to their major role in plant vigor, antioxidants in the context of long-term

dietary consumption can have antitumor, anti-inflammatory, neuroregenerative, blood lipid–lowering, and hypoglycemic effects on human health [12–16].

Anthocyanins and melanins are phenol-derived pigments that can give a blue, purple, or black color to barley kernels. Phenolic compounds are predominantly accumulated in grain envelopes—the husk, pericarp, and aleurone layer—and it has been shown that the amount of polyphenols correlates directly with the color intensity of the grain [17]. Anthocyanins accumulated in the aleurone layer and grain pericarp afford a blue and purple color, respectively [18]. The most common anthocyanin in purple barley is believed to be cyanidin 3-glucoside, followed by peonidin 3-glucoside and pelargonidin 3-glucoside, whereas in blue barley, delphinidin 3-glucoside usually dominates [19].

Melanins are another group of plant polyphenols, which, by accumulating in the grain pericarp and husk, can impart a black or brown color to seeds. Melanins are synthesized by enzymatic oxidation of simple phenolic precursors such as tyrosine, cinnamic acid derivatives, and catechol to quinones with subsequent polymerization [20]. Numerous studies of animal, fungal, and synthetic melanins have revealed their useful physicochemical properties including broadband absorption of UV–Vis–infrared radiation, electrically conductive properties including hybrid ionic-electron conductance, metal chelation, free-radical–scavenging activities, redox reversibility, and chemical versatility due to conjugation with other molecules. Moreover, growing interest in melanins is related to their good biocompatibility and biostability because a melanin is a polymer of natural origin [6,21,22]. Thus, due to its unique properties, it can be used in a broad range of biomedical and technological applications. Because seed envelopes of crops are usually byproducts of grain processing, they are a good and low-cost source of natural melanin. Decent potential of melanin from sunflower husks as a sorbent with high enterosorption efficiency has been demonstrated [23]. Moreover, there are uses of a melanin powder obtained from buckwheat husks as a food supplement with increased antioxidant capacity [24].

In barley, anthocyanin pigmentation of the grain pericarp is controlled by proteins encoded by two complementary genes, *Ant1* (MYB) and *Ant2* (bHLH), which are located on chromosomes 7H and 2H, respectively [25–28]. Together with transcription factor WD40, these proteins form the MBW complex, which regulates the expression of genes of the anthocyanin biosynthesis pathway in a tissue-specific manner [29]. Accumulation of blue anthocyanins in barley aleurone is controlled by the product of the *HvMyc2* gene, a paralog of *Ant2* [30]. Melanin biosynthesis in barley is under the monogenic control of *Blp1* located in the 0.8 Mb locus on chromosome 1H [31]. To date, the function of the *Blp1* gene is still unknown. Because the formation of the purple pigmentation (caused by anthocyanins) and the formation of black pigmentation (caused by melanins) are under independent genetic control, the pigments can be jointly accumulated in barley grains by combining the genes controlling their synthesis. Anthocyanin and melanin pigments together have been detected in Tartary buckwheat seeds, where anthocyanins appear first [32].

In the current study, to investigate whether the two types of pigments can accumulate together in the barley grain and how this combination affects plant pigmentation and developmental traits, a barley near-isogenic line (NIL) differing in *Ant1*, *Ant2*, and *Blp1* from parental lines was created by a marker-assisted approach and was compared with the parental lines, which accumulate only one type of pigment.

## 2. Materials and Methods

### 2.1. Plant Material and Marker-Assisted Creation of the NIL

To develop the barley NIL characterized by simultaneous accumulation of anthocyanins and melanins in seed envelopes (hereafter referred to as the black and purple line, BP), two NILs obtained in the spring with a cv. Bowman ('Bowman From Fargo', NGB22812, www.nordgen.org; accessed on 6 September 2021) genetic background were employed as parental lines. The first one is i:Bw*Blp1* (NGB20470, hereafter: BLP), which carries the *Blp1* locus on chromosome 1HL [33] and is characterized by melanin accumulation in hulls and in the grain pericarp. The second line is i:Bw*Ant1Ant2* (NGB22213, hereafter: PLP), which

is characterized by anthocyanin accumulation in the grain pericarp and leaf sheath and carries the complementary genes *Ant1* and *Ant2* on chromosomes 7HS and 2HL, respectively [33]. Microsatellite marker *Xgbms0184* linked with the *Blp1* locus [34] and intragenic PCR markers of genes *Ant1* and *Ant2* [35] were used for marker-assisted selection.

### 2.2. Phenotyping

Plants were grown at the Greenhouse Facility of the ICG SB RAS (Novosibirsk, Russia) under a 12 h photoperiod in a temperature range of 20–25 °C. Tillering (BBCH growth stage 21, BBCH-21), booting (BBCH-41), and heading (BBCH-51) stages were scored in 10 plants of each line. The time course of anthocyanin and melanin pigmentation in lines BLP and PLP and in the newly created BP line was examined starting from the milk stage of grain ripeness (BBCH-75). Every two days, 10 plants of each line were inspected, and the number of pigmented grains was recorded.

To determine the influence of *Blp1*, *Ant1*, and *Ant2* on the growth and yield parameters, the plants of cv. Bowman and of lines BLP, PLP, and BP were grown in an experimental field (Novosibirsk, N54.847102 and E83.127422) and on plots with regular watering in the ICG Breeding Genetic Complex. The plants were grown in 1 m rows, three rows for each line. The following parameters were scored for 20 plants in each row: dates of tillering, booting, and heading; the grain number and grain weight per plant; plant height; main spike length; and the spike number. Spike density was evaluated as the number of seeds per 4 cm of a spike from its base. Thousand-grain weight in each row was also calculated.

### 2.3. Anthocyanin Extraction and Measurement

Seeds of lines BLP, PLP, BP, and of cv. Bowman were picked at late milk (BBCH-77) and early dough (BBCH-83) stages of spike development. The husk and pericarp tissues were peeled and homogenized in liquid nitrogen. From mature seeds (whole grain), anthocyanins were extracted using a laboratory grain mill for homogenization. After that, 0.1 g of plant material was soaked in 500 μL of 1% HCl in MeOH. The mixture was incubated at 4 °C for 12 h and centrifuged at 12,000 rpm for 25 min at 4 °C. Absorbance was measured on a SmartSpecTMPlus spectrophotometer (Bio-Rad Laboratories, Inc., Hercules, CA, USA) at 530 nm. The total anthocyanin content was determined according to [36] and expressed in micrograms of cyanidin 3-glucoside (Cy-3-Glu) equivalents per gram of dry weight (DW) of sample material.

### 2.4. Cytological Analysis

To investigate the localization of melanin and anthocyanin pigments in grain envelopes, seeds of the studied lines were collected at the hard dough (BBCH-87) stage of spike development and frozen at −20 °C. Cytological analysis including cryosection preparation and assessment of visible pigments was performed according to previously described protocols [37] with identical equipment.

### 2.5. Statistical Analysis

Statistical analysis of the data was carried out in Statistica v. 6.1 software (StatSoft, Inc., Tulsa, OK, USA). Significance of differences between parameters of the lines was assessed by the *t* test, with $p < 0.05$ indicating significance. The nonparametric Kruskal–Wallis *H* test (ANOVA) was performed for determining the influence of the pigments (melanin and anthocyanin) on yield components of the barley NILs. Correlations between the parameters were evaluated by means of Spearman's rank correlation coefficients.

## 3. Results

### 3.1. Construction of the NIL

F$_2$ plants were obtained via a crossing of the BLP line (genotype *ant1ant1ant2ant2 Blp1Blp1*) and PLP line (*Ant1Ant1Ant2Ant2 blp1blp1*). The selection of the desired plants was carried out in three steps (Figure 1): first, F$_2$ plants with the homozygous *ant1* allele,

which causes the absence of anthocyanin pigmentation in the leaf sheath, were excluded from further analysis. Next, DNA was isolated from the remaining $F_2$ plants (24 total), and marker-assisted selection was conducted using intragenic PCR markers for genes *Ant1* and *Ant2* as well as SSR-marker *Gbms184* linked with the *Blp1* locus (Figure S1). As a result, four plants of the *Ant1Ant1Ant2Ant2* genotype were selected, two of which had a heterozygous dominant *Blp1* allele. The $F_3$ generation was obtained from the selected plants, and six plants with genotype *Ant1Ant1Ant2Ant2Blp1Blp1* (characterized by the simultaneous accumulation of anthocyanins and melanins in the grain envelopes) were selected. Next, the phenotype stability of the obtained $F_4$ and $F_5$ plants was tested. For further analysis, the progeny of one of these plants were used and designated as either the i:Bw*Ant1Ant2Blp1* or BP line.

### 3.2. Comparative Analysis of the Pigmentation Time Course

It was found that all lines under study went through key stages of development at the same time: they reached tillering stage by 8–11 days after sowing and booting and heading stages by 35–36 and 40–43 days, respectively, after sowing. The time course of spike pigmentation in the NILs was also examined. It was noticed that the emergence of anthocyanin pigmentation in the PLP and BP lines began at the late milk stage of grain ripeness (50–52 days after sowing), whereas by day 55 after sowing, the seeds of lines PLP and BP proved to be fully pigmented with anthocyanin. On the other hand, the appearance of melanin pigments in the BLP and BP lines began only at 55–57 days after sowing when the plants reached the early dough stage of grain ripeness. Therefore, accumulation of anthocyanins and accumulation of melanins in barley are separated in time in both the parental lines (BLP and PLP) and in the hybrid line BP. Phenotypes of cv. Bowman and lines BLP, PLP, and BP are presented in Figure 2.

### 3.3. Cytological Analysis of Grain Pigmentation

Seeds of lines BLP, PLP, BP, and of cv. Bowman at the hard dough stage of spike ripening were inspected by light microscopy (Figure 3). In the Bowman cultivar, all layers including aleurone, pericarp, and husk were clearly visible, but none of them contained pigments. In the pericarp and husks of the BLP line, a large number of round-shaped brownish-black structures with sizes of 2–5 μm were revealed both in the pericarp and husks layers. Brownish-red rod-shaped pigmented structures with 5–10 μm widths and 40–100 μm lengths were found in the pericarp of the PLP line, while pigments were not observed in husks. In the hybrid line BP, round-shaped brownish-black structures with sizes of 2–5 μm were observed in husks similar to the BLP line, but were absent in the pericarp. At the same time, in the pericarp of the BP line, two types of rod-shaped pigmented structures were observed: brownish-red, similar to PLP, and brownish-black, unique rod-shaped pigmented structures that were not observed in the parental lines. None of the pigments were observed in the aleurone layer of all lines under investigation.

### 3.4. The Anthocyanin Content of the Grain of NILs

Anthocyanin pigments were extracted from the husk and grain pericarp and quantified in the lines under study at late milk and early dough stages of spike development. It was observed that the anthocyanin content of the BLP grain was as low as that of unpigmented cv. Bowman at both stages (Figure 4A). In the PLP grain, the anthocyanin content was $152.74 \pm 4.12$ μg/g (mean $\pm$ SD) at the late milk stage, while in the BP grain, it was ~3.8-fold higher ($578.76 \pm 64.88$ μg/g) in comparison with the parental PLP line. At the early dough stage, the anthocyanin concentration was $320.25 \pm 20.52$ μg/g in the PLP grain, while in the BP grain, it was almost the same as that at the previous stage ($566.48 \pm 39.63$ μg/g). In extracts from whole mature grains, the concentration of anthocyanins was not significantly different between lines PLP and BP (Figures 4B and S2).

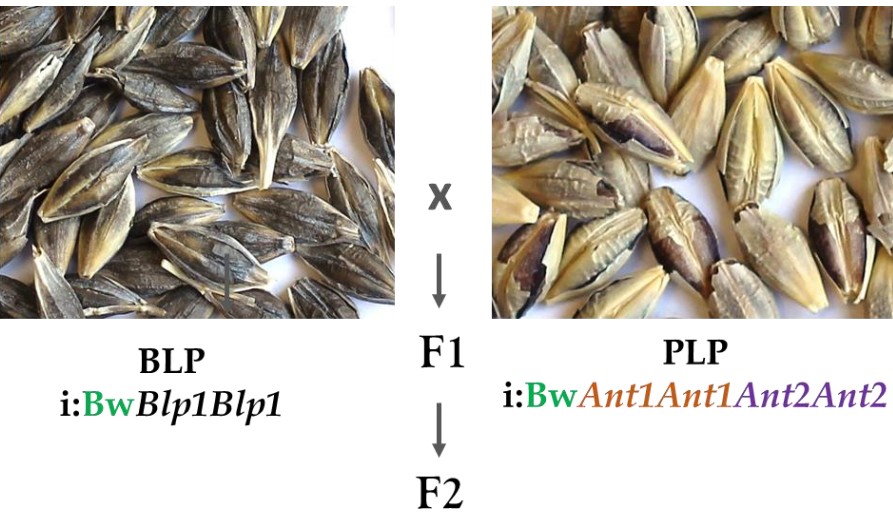

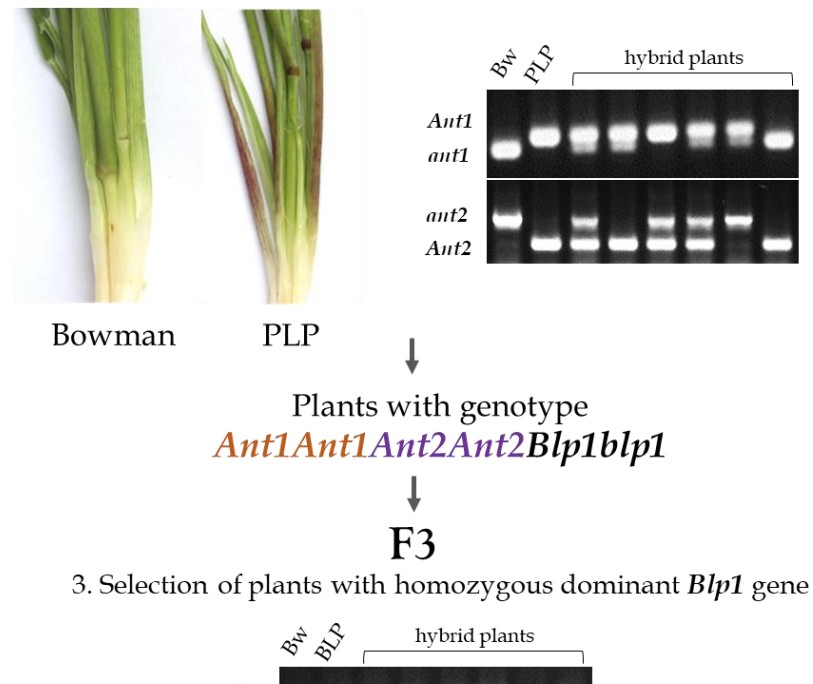

**Figure 1.** The strategy for marker-assisted selection of the barley NIL carrying dominant alleles of genes *Ant1*, *Ant2*, and *Blp1* in the Bowman genetic background. NILs i:Bw*Ant1Ant2* and i:Bw*Blp1*, characterized by purple and black pigmentation of the grain, served as donors of the dominant alleles of *Ant1*, *Ant2*, and *Blp1*, respectively. The three-step selection of homozygous $F_3$ plants for the target genes was carried out on the basis of leaf sheath pigmentation assessment (1), donorlike homozygous diagnostic markers of genes *Ant1* and *Ant2* (2), and an SSR-marker linked with the *Blp1* gene (3). The selected plants homozygous for the target genes were propagated and checked for phenotype stability in generations $F_{4-5}$. One plant from a group devoid of segregation of the pigmentation traits was selected as NIL i:Bw*Ant1Ant2Blp1*.

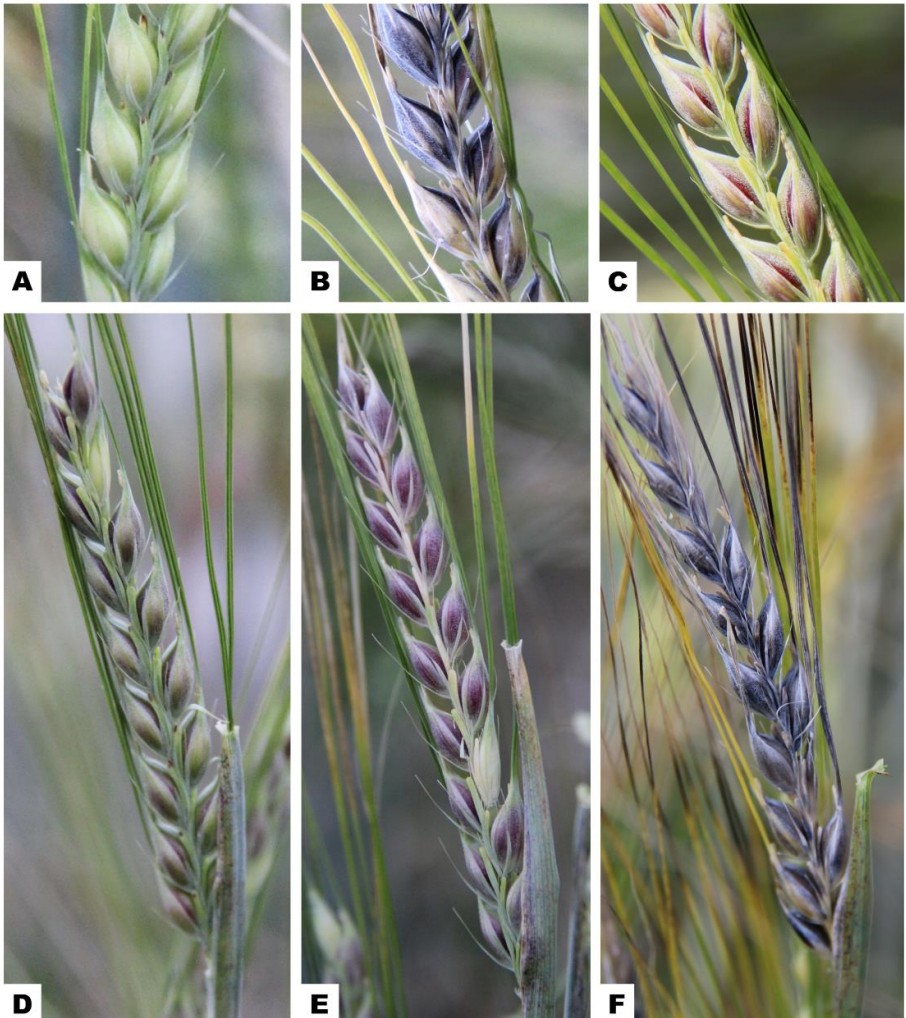

**Figure 2.** Spikes of the barley lines under study: (**A**) cv. Bowman, (**B**) BLP, and (**C**) PLP. The BP line at late milk (**D**), early dough (**E**), and hard dough (**F**) stages of spike development.

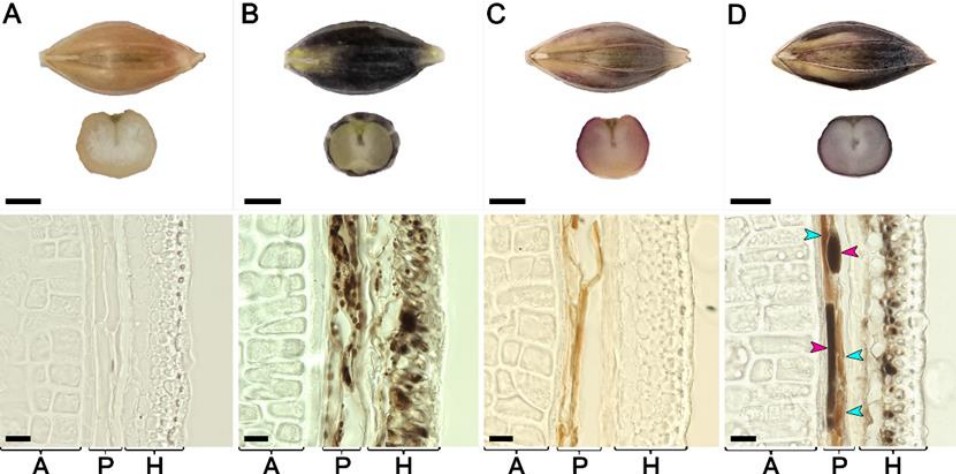

**Figure 3.** Seeds and cross-sections of the seeds produced by cv. Bowman (**A**) and lines BLP (**B**), PLP (**C**), and BP (**D**) sampled at the hard dough stage. A: aleurone layer, H: husk, P: pericarp. Turquoise arrowheads point to brownish-red rod-shaped structures, and magenta arrowheads indicate brownish-black rod-shaped structures. Scale bars for the seeds are 2.5 mm, and for the micrographs, 20 μm.

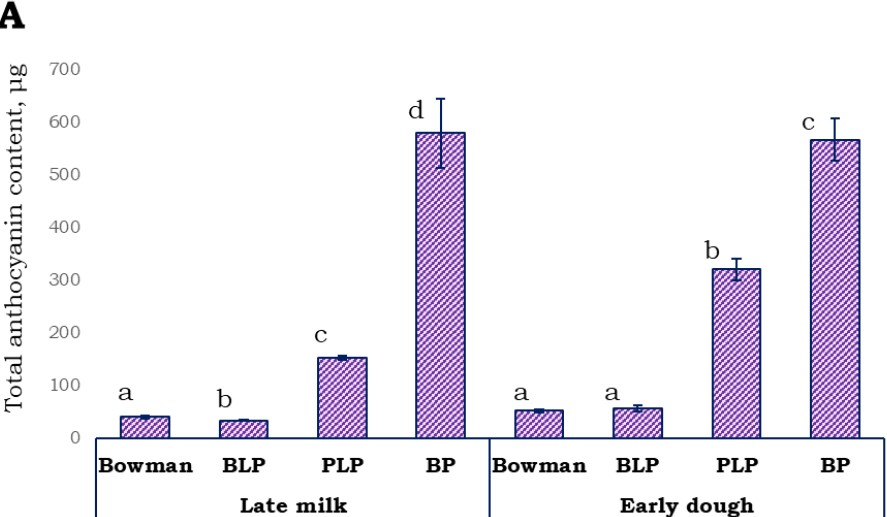

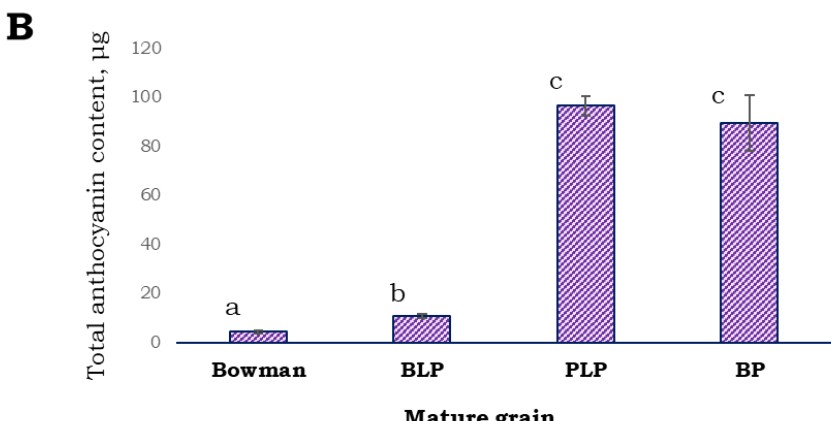

**Figure 4.** The anthocyanin content expressed in Cy-3-Cly equivalents per gram of DW in the grain pericarp and husk at the late milk and early dough stages of spike development (**A**) and in the whole mature grain (**B**). Different letters indicate statistically significant differences between the lines (*t* test, $p < 0.05$).

### 3.5. Evaluation of Yield Components of the NILs

To reveal the effects of genes *Ant1*, *Ant2*, and *Blp1* on yield components, a comparative phenotyping analysis was performed on the NILs (Table 1). The lines grown in the field were not significantly different in height, spike number per plant, spike length, and the number of seeds per plant. In contrast, the BLP line had higher spike density in comparison with PLP (12.53 ± 0.38 and 11.66 ± 0.08, respectively) and had a significantly greater grain weight per plant than cv. Bowman (3.05 ± 0.28 and 2.17 ± 0.35 g, respectively). Moreover, thousand-grain weight was significantly greater in lines BLP and BP (which are characterized by melanin accumulation) than in the PLP line (containing anthocyanin in the grain pericarp).

Plants grown under less severe conditions (i.e., on plots with regular watering) had greater values of all the tested parameters in all the lines. The differences seen between the lines in the field were not confirmed on the plots. The lines with anthocyanins in grains were taller than the melanin-accumulating BLP line and had significantly greater spike length than the BLP line and cv. Bowman. There were no significant correlations between the yield characteristics of the lines under the field and plot conditions with the exception of the plant height parameter (Table S1).

**Table 1.** Characteristics of the lines with different combinations of anthocyanin and melanin pigments. Different letters within a row indicate statistically significant differences between the lines (*t* test, $p < 0.05$).

| Trait\Line | Bowman | BLP | PLP | BP |
|---|---|---|---|---|
| | | Field | | |
| Plant height, cm | 75.15 ± 2.66 | 75.03 ± 0.80 | 76.38 ± 1.16 | 76.12 ± 1.74 |
| Spike number per plant | 3.63 ± 0.67 | 4.47 ± 0.48 | 3.93 ± 0.54 | 3.57 ± 0.58 |
| Spike length, cm | 5.25 ± 0.18 | 5.52 ± 0.06 | 5.7 ± 0.33 | 5.62 ± 0.28 |
| Spike density | 11.58 ± 0.55 [ab] | 12.53 ± 0.38 [a] | 11.66 ± 0.08 [b] | 11.93 ± 0.38 [ab] |
| Number of seeds per plant | 43.92 ± 6.92 | 58.87 ± 7.48 | 51.32 ± 7.09 | 47.28 ± 7.62 |
| Grain weight per plant, g | 2.17 ± 0.35 [a] | 3.05 ± 0.28 [b] | 2.42 ± 0.30 [ab] | 2.40 ± 0.39 [ab] |
| Thousand-grain weight, g | 49.43 ± 0.60 [ad] | 51.88 ± 2.08 [ad] | 47.22 ± 0.69 [b] | 50.82 ± 0.60 [c] |
| | | Plots | | |
| Plant height, cm | 78.67 ± 1.94 [ab] | 76.47 ± 1.10 [a] | 80.78 ± 1.96 [b] | 79.99 ± 3.49 [ab] |
| Spike number per plant | 4.8 ± 0.40 | 4.95 ± 0.80 | 5.0 ± 0.52 | 5.66 ± 0.93 |
| Spike length, cm | 6.72 ± 0.25 [a] | 6.57 ± 0.08 [a] | 7.38 ± 0.16 [bc] | 7.45 ± 0.33 [c] |
| Spike density | 10.28 ± 0.18 | 10.47 ± 0.08 | 10.2 ± 0.40 | 10.42 ± 0.30 |
| Number of seeds per plant | 65.57 ± 4.64 | 65.85 ± 10.36 | 72.43 ± 11.45 | 76.41 ± 16.96 |
| Grain weight per plant, g | 3.57 ± 0.27 | 3.55 ± 0.61 | 3.90 ± 0.53 | 4.13 ± 0.92 |
| Thousand grain weight, g | 54.46 ± 0.64 | 53.8 ± 0.85 | 54.03 ± 1.95 | 55.04 ± 0.35 |

One-way ANOVA on ranks showed that the presence of anthocyanin or melanin may exert some positive effects on growth parameters of the plants. It was observed that anthocyanins in these barley lines could affect linear parameters of the plants such as plant height and spike length, whereas the presence of melanin could increase spike density (Table S2), as confirmed by observations both in the field and on the plots.

## 4. Discussion

The yellow color of the barley grain is common, and most of its released cultivars are yellow-grained, while pigmented barleys such as purple, blue, and black cultivars also exist and have recently received much attention as new sources of functional-food ingredients [38,39]. The different types of grain pigmentation (purple, blue, and black) are controlled by different genes, which may be present in the genome in a dominant state simultaneously and cause combined accumulation of the pigments. For instance, this principle has been demonstrated for blue and purple anthocyanins; moreover, their combination in one genotype looks like a promising strategy to increase the nutritional value of the barley grain [40]. In the case of melanin, the tracking of this pigment, together with anthocyanins in the mature grain, is problematic because black melanins accumulating in outermost husk tissue mask the purple and blue anthocyanins that accumulate more deeply in the grain. There are reports of the presence of anthocyanin pigments in grains before their maturity. For instance, in highland barley, a purple color may appear in the spike before grain maturity, but disappears or turns into a dark color (gray or black) when the grain matures [41]. Although the chemical nature of the dark pigmentation mentioned in that study and genotypes of the samples were not determined, one can assume that melanins mask anthocyanin pigments accumulating earlier in development.

In the current study, a NIL that accumulates both anthocyanin and melanin pigments in the grain was designed via a marker-assisted approach by targeted combining (in one genotype) the dominant alleles of the genes (*Ant1*, *Ant2*, and *Blp1)* that control the synthesis of these pigments. Clear-cut genetically controlled accumulation of anthocyanins and melanins in one grain was observed in the resultant line (BP) during spike development. As in highland barleys, the emergence of anthocyanin pigmentation and that of melanin are temporally separated in BP, and the production of anthocyanin pigments begins at an earlier stage of spike maturation.

The round-shaped brownish-black structures with sizes of 2–5 μm observed in the BLP and BP lines were previously identified as melanoplasts, melanin-accumulating plastids [37]. The brownish-red rod-shaped pigmented structures observed in PLP and BP were identified as anthocyanin containing vacuoles on the basis of the known cellular localization of these pigments [8,42]. The unique brownish-black rod-shaped structures were observed only in pericarp of BP line. Taking into account that clearly visible melanoplasts were not observed in the pericarp of the BP line, it can be assumed that brownish-black rod-shaped structures resulted from the fusion of melanoplasts containing melanins and vacuoles containing anthocyanins. The assumption is supported by observation of plastid fusion involved into melanin synthesis with vacuoles in *Piptocarpha axillaris* (Less.) [43]. This allowed us to conclude that in plant tissues, melanin synthesis takes place in plastids and could be accumulated in both the plastids and vacuoles.

Quantitative estimation of the total anthocyanin content in this set of NILs revealed that at late milk and early dough stages, the BP pericarp contained threefold more anthocyanins than the pericarp of parental line PLP. Nonetheless, in mature seeds, the amounts of anthocyanins were similar between BP and PLP, and the observed differences between the lines could be explained by environmental factors. From these data, one can conclude that the *Blp1* gene positively affects the anthocyanin content at some developmental stages. Previously, an overlap of the biosynthetic pathways leading to melanins and proanthocyanidins (which share some biosynthesis stages with anthocyanins) at the initial steps of phenylpropanoid biosynthesis was hypothesized after examination of *bks* (*black seed*) and a series of *anthocyaninless* mutants of tomato [44]. Moreover, the gene affecting both melanin synthesis in seeds and anthocyanin synthesis in flowers has been identified [*ItIVS* (bHLH)] in morning glory [45]. All of these findings, in combination with those presented here, are obvious evidence of some interactions between the melanin and anthocyanin pathways at the molecular-genetic level; however, precise mechanisms of these interactions have not been previously investigated.

The possibility of obtaining barley genotypes combining anthocyanins and melanins as demonstrated here (both types of pigments have high nutritional potential [7,38,46]) as well as the positive impact of the *Blp1* gene on anthocyanin concentration in the grain are suggestive of a promising strategy for creating polyphenol-rich barley genotypes for functional foods. On the other hand, the accumulation of pigments in seed envelopes is believed to negatively affect the productivity of plants and their yield. For example, in purple rice, a yield reduction in comparison with uncolored rice has been documented that could be explained by a lower photosynthesis rate in pigmented tissues [47]. This negative influence is debatable because anthocyanins as strong antioxidants can provide some protection of the photosynthetic apparatus from oxidative stress [48,49]. Our new line, together with the parental ones, represents a precise genetic model for researching the effects of the genes controlling the accumulation of different types of pigments and their combinations on plant growth and development. The comparison of these NILs in terms of yield components here suggests that the observed differences between the barley NILs are caused by environmental factors. Nevertheless, according to ANOVA, it can be supposed that the presence of pigments can have a positive impact on the linear parameters of plants such as plant height, spike length, and density, as confirmed by our observations both in the field and on the plots. In one report, a comparison of yield parameters among several wheat NILs carrying different combinations of genes *Pp3* and *Pp-1* (purple pericarp) orthologous to barley *Ant2* and *Ant1*, respectively, revealed that the differences in the yield among the lines are also determined by environmental factors [50]. The black color of the barley kernel is believed to be an adaptive trait under harsh conditions such as the cold, drought, and high altitudes [51,52]. Black barleys are thought to possess early growth vigor, cold tolerance, greater height, and earlier maturity, which could give them an advantage under dry conditions [9]. Overall, our results did not show any negative effects of the genes determining pigment accumulation on plant growth, and some traits were even positively affected by the genes. The observed advantages of the BP line in the pigment content over

the other lines—containing only one type of pigment—and the absence of any obvious negative effects of the corresponding genes may be useful for breeding programs aimed at designing barley varieties enriched with polyphenolic compounds applicable as promising ingredients of functional foods.

## 5. Conclusions

This comparative study of the NILs with different combinations of anthocyanins and melanins in grain envelopes revealed that the accumulation periods of these pigments are temporally separated, and this observation can be utilized as a phenotyping marker for the prediction of the pigment composition of barley grains. Even though anthocyanins and melanin pigments have different natures, it can be proposed that the *Blp1* gene can alter the anthocyanin content of the barley grain. The hypothesis of a suppressive effect of pigment accumulation in grain envelopes on the yield and plant productivity was not confirmed in this study; the observed differences between NILs are explained by environmental factors. Thus, further investigation of barley genetic diversity and its adaptation to specific environments in conjunction with a relevant breeding methodology can contribute to the creation of barley varieties with advanced technological applications.

**Supplementary Materials:** The following are available online at https://www.mdpi.com/article/10.3390/agronomy12010112/s1, Figure S1: Electropherograms illustrating the PCR markers employed for the selection based on genes *Ant1*, *Ant2*, and *Blp1*, Figure S2: Anthocyanin levels in whole grains grown in the field and on plots, Table S1: Spearman's rank coefficients of correlation between the barley lines' linear parameters and yield components between NILs from the field and plots, Table S2: Effects of anthocyanin and melanin accumulation as revealed by ANOVA.

**Author Contributions:** Conceptualization, O.S. and E.K., Investigation, A.G., T.K. and S.M.; Formal analysis, A.G.; Writing—original draft, A.G. and O.S., Writing—review and editing, S.M. and E.K. All authors have read and agreed to the published version of the manuscript.

**Funding:** The study was funded by the Russian Foundation for Basic Research, project number 20-316-80016. The cultivation of the barley plants at the Greenhouse Facility was supported by ICG project 0259-2021-0012.

**Acknowledgments:** We thank Nikolai Shevchuk for the linguistic advice and proofreading of the manuscript.

**Conflicts of Interest:** The authors declare no conflict of interest.

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
