# Peer review of "Effects of Combining the Genes Controlling Anthocyanin and Melanin Synthesis in the Barley Grain on Pigment Accumulation and Plant Development"

_agronomy, doi:10.3390/agronomy12010112_

Round 1

Reviewer 1 Report

Dear editor and authors!

The article takes into consideration an interesting and highly current problem of the regulation of pigment metabolism during plant development. The establishment of temporal biosynthesis pattern of antocyanins ond melanins in barley is thought-provoking, especially in combination with gene mapping and yield impact analysis. I believe the results of this study may find application in both agronomy and biotechnology.

The observation that lines with higher melanin content tend towards higher grain weight is worth investigating further, in my opinion.

I find the article very interesting, well written and I think it will be to the liking of readers of Agronomy, there are however some very minor improvements that would be beneficial to include before publication:

Comments

Lines:

61-61: “,,, hybrid ionic-electronic conductance…” It is slightly unclear. Do the Authors imply that melanins take part in signal transduction? Or something else entirely? It should be clarified.

243: “…with the exception…”

Author Response

Dear Reviewer,

Thank you very much for your critical reading and valuable comments on our manuscript. We appreciate your high evaluation of our study and did our best to improve it according to your recommendations.

Line 61: Melanin is an electrically conductive material which is sustaining a hybrid ionic-electronic behavior where the electronic contribution depends on the presence, extent and the redox properties  of the delocalized aromatic systems, while the ionic part is largely dictated by the hydration level of the material (Mostert et al., 2012, Role of semiconductivity and ion transport in the electrical conduction of melanin, doi: 10.1073/pnas.1119948109; Wünsche et al., 2015, Protonic and Electronic Transport in Hydrated Thin Films of the Pigment Eumelanin, https://doi.org/10.1021/cm502939r). We have clarified the conductive properties of melanin in the manuscript and added the references mentioned above.

Line 243: Fixed in the text of manuscript.

Please see the attachment with revised manuscript.

Best regards,

Anastasiya Glagoleva

Reviewer 2 Report

Dear authors,

This is an interesting and well written manuscript. I really enjoyed reading this research and the dataset is good.

  1. All key elements are present.

  1. The title clearly describes the article.

  1. The abstract content clearly reflects the entire content of the article.

  1. In the introduction paragraph the authors clearly estate the problem investigated. Also, the purpose of the study is specified.

  1. The authors accurately explain the field experiments.

  1. Data are well presented

Author Response

Dear Reviewer,

Thank you very much for your revision and high evaluation of our work.

Best regards,

Anastasiya Glagoleva

Reviewer 3 Report

sentence between row 51a-53 ref  [18].

recent reference such folowing bring more recent information about BLp1  inheritance and linked markers: 

https://www.frontiersin.org/articles/10.3389/fpls.2017.01414/fullEffects of the Blp1 locus, which controls melanin accumulation in the barley ear, on the size and weight of seeds : 30901/2227-8834-2021-2-89-95

question: did yu try to soak seeds on water or concentrated HCL as usually is the test that we do for Anthocyanin pigment on seeds , if it desapear it means its anthocyanin , the melanin are morelikly tannin they should not be diluted so for the seeds combining both traits it my be the test that can confirm that the seed is a comination of the two traits .?!

Author Response

Dear Reviewer,

Thank you for your valuable input and we do our best to clarify your question.

In our study, we didn't use concentrated HCl to test the anthocyanins presence in barley grain. The anthocyanins extraction procedure which includes soaking of grains in methanol with 1% HCl was used. Using this approach, it is possible to detect the presence of anthocyanins both in samples that contain only anthocyanins and in samples that accumulate anthocyanins and melanins. You are right that melanins do not dissolve in acids, as they do in alcohols, and it is likely that the approach you proposed can be useful for identifying pigments. However, it will be most reliable to carry out the extraction of pigments according to standard protocols, such as mentioned above HCl/methanol extraction for anthocyanins (Abdel-Aal et al., 1999, doi:10.1094/CCHEM.1999.76.3.350) and alkaline extraction for melanins (Sava et al., 2001, A novel melanin-like pigment derived from black tea leaves with immuno-stimulating activity, DOI:10.1016/S0963-9969(00)00173-3).

Lines 51-53. The reference regarding to distribution of pigments in grain envelopes was added.

Please see the attachment with revised manuscript.

Best regards,

Anastasiya Glagoleva

Reviewer 4 Report

the authors must add statistical analysis at the end of materials and methods and must add the letters of statistical analysis on figures

the supplemented figures must add to the paper instead of the figures added

the English must be improved

the references must be updated in 2021

Author Response

Dear Reviewer,

Thank you your critical reading and valuable comments.

‘the authors must add statistical analysis at the end of materials and methods and must add the letters of statistical analysis on figures’

The description of statistical analysis was placed in the end of Materials and Methods part. The letters of statistical analysis were added on Figure 4 and Figure S2.

‘the supplemented figures must add to the paper instead of the figures added’

We had taken into consideration your comment, but illustrations given in the text of the manuscript fully reflect the content of the article, and figures in the Supplementary provide some additional information, which, nevertheless, is not key to understanding of the work.

‘the English must be improved’

The English was previously improved by professional English editor Dr. Nikolai Shevchuk who mentioned in the Acknowledgments.

‘the references must be updated in 2021’

Probably we did not fully understand your comment, but the references in the text are formatted according to Agronomy Reference List instruction.

Please see the attachment with revised manuscript.

Best regards,

Anastasiya Glagoleva

Round 2

Reviewer 4 Report

the authors made all corrections